# Conversion of a Single-Frequency X-Band EPR Spectrometer into a Broadband Multi-Frequency 0.1–18 GHz Instrument for Analysis of Complex Molecular Spin Hamiltonians

**DOI:** 10.3390/molecules28135281

**Published:** 2023-07-07

**Authors:** Wilfred R. Hagen

**Affiliations:** Department of Biotechnology, Delft University of Technology, Building 58, Van der Maasweg 9, 2629 HZ Delft, The Netherlands; w.r.hagen@tudelft.nl

**Keywords:** EPR, ESR, broadband, strip line, wire micro strip, metal complex, metalloprotein, free radicals

## Abstract

A broadband EPR spectrometer is an instrument that can be tuned to many microwave frequencies over several octaves. Its purpose is the collection of multi-frequency data, whose global analysis affords interpretation of complex spectra by means of deconvolution of frequency-dependent and frequency-independent interaction terms. Such spectra are commonly encountered, for example, from transition-metal complexes and metalloproteins. In a series of previous papers, I have described the development of broadband EPR spectrometers around a vector network analyzer. The present study reports on my endeavor to start from an existing X-band spectrometer and to reversibly re-build it into a broadband machine, in a quest to drastically reduce design effort, building costs, and operational complexity, thus bringing broadband EPR within easy reach of a wide range of researchers.

## 1. Introduction

EPR spectra generally change shape with changing microwave frequency, *ν*, because they are monitors of multiple electron interactions, some of which are linear in *ν*, while others are independent of *ν*. Linear interactions are the electronic Zeeman interaction and also the g-strain resulting from molecular conformational distributions. Examples of independent interactions are electron–nuclear hyperfine interactions, zero-field interactions in high-spin systems, and dipolar interactions between paramagnetic centers. Collection of data at multiple frequencies, and their subsequent global analysis, increases our chances for a meaningful spectral interpretation in terms of a unique spin Hamiltonian. A common application of this philosophy is multiple high-frequency EPR for the resolution of g matrices with small anisotropy and for the detection of systems with zero-field splitting significantly greater than the standard X-band quantum of ca 0.3 cm^−1^ [1,2]. A similar, though less common, approach at lower frequencies (of order X-band and below) is indicated for resolution and interpretation of complex patterns of hyperfine and/or dipolar interaction.

Over the last decade, I have developed versions of low-frequency broadband EPR as a practical solution for multi-frequency studies around and below X-band [3,4,5,6]. Commercial low-frequency EPR spectrometers have been available previously; however, their application has always been limited due to their single-frequency nature: each additional frequency requires an additional investment of order 200 k€ (or k$), and each step in frequency requires the interchange of single-mode resonators and associated cryogenics. My broadband spectrometers were conceived as single stations with a resonator circuit that is tunable to many different frequencies, without the need to change resonators and cooling systems.

In spite of the obvious advantages of the broadband system, it has yet to show wider distribution, which I understand from discussions with interested colleagues is rooted in perceived fear for excessive costs of construction and for excessive complexity in construction and operation. I have therefore re-thought the concept of broadband EPR hardware, and I have now designed a simple and affordable (<15 k€) conversion kit intended as an extension of commercial standard X-band spectrometers (I have used a Bruker EMXplus) into an easily operable general purpose broadband spectrometer for experiments in the approximate frequency range of 0.1–18 GHz. The kit consists of a few readily commercially available microwave devices (source, circulator, amplifier, power meter, detection diode) and a few items that can be easily constructed at low cost and without special skills (resonator, modulation coils). Assembly of the units into a broadband bridge with SMA cables is a ten-minute job, and so is the subsequent implementation of the bridge into the X-band spectrometer. Open-source software to operate the broadband bridge is provided in the form of a simple, modifiable LabVIEW program.

## 2. Basic Theory

Since 1947, in EPR spectroscopy, the microwave frequency standard is X-band (circa 9–10 GHz) applied to a single-frequency cavity resonator [7]. ‘Standard’ here means that for many molecular systems, X-band EPR provides an approximate optimum in signal-to-noise ratio, from a balancing of increased sensitivity with frequency due to the Boltzmann distribution over states, and decreased sensitivity with higher frequency due to intrinsic hardware properties such as the noise characteristics of detectors. In practice, this has led to a situation in which the majority of EPR experiments are conducted with commercial X-band spectrometers, and only when the results are insufficient for unequivocal interpretation, additional experiments are carried out in different frequency bands, whereby each additional frequency typically requires the laborious and costly procedure of employing an additional spectrometer. Broadband EPR spectroscopy represents a different approach in which the single-frequency cavity is abandoned in favor of a resonator that can be tuned to many different frequencies over many octaves, all in a single spectrometer. In practice, it is useful to operate the broadband spectrometer next to a standard X-band machine because the sensitivity of the former is significantly lower than that of the latter, although this difference may be less at frequencies outside the X-band [4,5].

Microwaves can be efficiently (that is, with minimal loss of energy) transmitted in a number of ways. In the standard spectrometer, this is carried out by means of waveguides of specific cross dimensions for the X-band. In the broadband spectrometer, transport is via a combination of coaxial cable and, for the resonator, a specific variant of strip-line technology: wire micro strip circuitry, whose basic geometry is illustrated in Figure 1. Its four design parameters, *d* (wire diameter), *h* (height of wire above the ground plate), *b* (extension of sample above the wire), and *ε_R_* (dielectric constant or relative electric permittivity of the sample), determine its characteristic impedance *Z*_0_ via intermediate parameters w (height of square wire equivalent to round wire of diameter d), *ε_eff_* (effective permittivity), and *ε_emb_* (embedded permittivity) as follows [4]:(1)w=d/1.1803
(2)$$\varepsilon_{eff}=\left(\frac{\varepsilon_{R}+1}{2}\right)+\left(\frac{\varepsilon_{R}-1}{2}\right)\sqrt{\frac{w}{w+12h}+X}$$
(3)$$X=0.04{\left(1-\frac{w}{h}\right)}^{2}\;\mathrm{for}\;w/h<1,\;\mathrm{and}\;X=0\;\mathrm{for}\;w/h\geq 1$$
(4)$$\varepsilon_{emb}=\varepsilon_{eff}e^{-2b/h}+\varepsilon_{R}\left(1-e^{-2b/h}\right)$$
(5)$$Z_{0}=\frac{337}{2\pi \sqrt{\varepsilon_{R}}}{cosh}^{-1}\left(\frac{2h+d}{d}\right)$$
in which 337 Ω is the characteristic impedance of air. With *ε_R_* given for a particular sample, the parameters *d*, *h*, and *b* are adjusted to afford a characteristic impedance as close as possible to *Z*_0_ = 50 Ω. This is the universally used value for all components in microwave circuitry, to guarantee lossless connection. A simple calculator of *Z*_0_ from *ε_R_*, *d*, *h*, and *b* is given in Appendix A. 

For example, for a compound with *ε_R_* = 3.15, the values *d* = 0.25 mm, *h* = 0.15 mm, and *b* = 0.1 mm give *Z*_0_ = 49.7 Ω. Also, the *Z*_0_ value is found to be not very sensitive to the value of *b* as long as *b* > 0. This leads to a resonator design (geometric details in Section 3) consisting of a metal ground plate covered with four layers of isolating acrylate tape (thickness 50 μm), in which a rectangular sample compartment is cut out that is surrounded by tight windings of metal wire of 0.25 mm diameter such that the isolator is compressed to circa 0.15 mm. When the compartment is filled with sample that extends somewhat above the wire, the resulting characteristic impedance will be approximately 50 Ω. Ice has an *ε_R_* = 3.15 [8], and dilute frozen solutions will not have a very different dielectric constant, so this cell is appropriate, e.g., for frozen aqueous solutions of metal complexes and metalloproteins. Moreover, since powders (microcrystals) of inorganic salts typically have similar *ε_R_* values, broadly in the range 2–5 [9,10], the cell design is also an acceptable compromise for broadband EPR on many inorganics.

## 3. Results

### 3.1. Global Description of the Conversion

A schematic overview of the broadband spectrometer is given in Figure 2. Parts of the existing X-band spectrometer are represented in gray; they can be used without modification to control the magnetic field and to encode/decode the EPR signal with 100 kHz field modulation in the signal-channel unit. Also, the proprietary software for operation of the machine is retained (with slightly modified instructions; see below). The X-band bridge is not used and is replaced with a broadband bridge whose elements are represented in pink. All these elements are readily commercially purchasable from multiple vendors. The new bridge is controlled with a dedicated PC (Windows). The X-band cavity (with build-in modulation coils) and waveguide is demounted from the X-band bridge and is removed. It is replaced with items represented in yellow to indicate that they are to be constructed in-house: the multi-frequency resonator (alias sample holder) and a pair of modulation coils providing sufficient space to encompass the resonator and an He/N_2_ flow dewar. The latter is omitted from the drawing for clarity and is described later.

All microwave, or RF, elements of the bridge are interconnected with blue coaxial cable (e.g., RG402) using SMA male and female connectors. The broadband RF diode detecting the EPR signal has a video output with SMA connector, which is converted to BNC using an adaptor. It is connected to the BNC signal channel cable previously disconnected from the X-bridge, whose other end still connects to the phase-sensitive detector in the signal-channel unit of the X-band spectrometer.

### 3.2. Details of Purchasable Parts

As a microwave source, I use a SynthHD PRO (ca 3500 €) from Windfreak Technologies (New Port Richey, FL, USA), which is a tunable and scannable source from 10 MHz to 18 GHz with ca +18 dBm maximal output power (the ‘m’ in ‘dBm’ defines the power as absolute: 0 dBm corresponds to 1 milliwatt; 18 dBm is 80 mW). Other vendors carry items of similar specifications. The source is controlled by the PC via a USB cable. Note that a microwave source is an active RF device (it consumes energy from a power supply), which means (1) that it produces some heat that must be diverted to a sink such as an aluminium ground plate, and (2) that it is a static-sensitive device, which must be handled with care (ground operator hands) when being connected to other RF circuity. 

The circulator accepts microwaves at its port-1, transfers them to port-2 for delivery to the sample, and then power reflected back from the resonator into port-2 is directed to port-3 for detection. Circulators are passive devices and not vulnerable to external disturbances (except for very strong local magnetic fields). The circulator is probably the only item on our shopping list that is not obtainable in a form that covers the full frequency range that we desire. Over the years, I created a wide collection through direct purchase, second-hand purchase, and removal from old equipment. For most practical purposes I suggest purchasing three circulators to cover the ranges 1–2, 2–6, and 6–18 GHz, such as the broadband coaxial SMA circulators (ca 3 × 400 €) from UIY (Shenzhen, PRC). Other vendors carry items of similar specifications. Exchanging the circulators to cover different frequency bands is a minor inconvenience compared to the exchange of resonators and dewars required with single-frequency bridges. The exchange may be automated by building a switching unit with multiple electronic RF switches.

The amplifier must boost the weak reflection signal from the resonator to a level that is optimal for detection by the RF diode. Under-amplification leads to poor signal-to-noise ratios; over-amplification may result in destruction of the diode. The balancing act is taken care of by the software within some limitations imposed on the amplifier. Amplification should typically be of order 20 dB, and saturation (in RF language: the *P_1dBm_* point) should preferably occur below the maximally allowed input power level of the diode. Single amplifiers with these specifications and covering a frequency range of 1–18 GHz or more are available at a price. Practically, I suggest purchasing two amplifiers, e.g., to cover the frequency ranges 0.5–8 and 6–18 GHz such as the coaxial broadband amplifiers (together ca 600 €) from Mini-Circuits (Brooklyn, NY, USA). Amplifiers are active devices and must be connected to a cooling sink.

The amplifier is followed by a SPDT (single pole double throw) RF switch for DC-18 GHz (ca 500 € new; ca 50 € from eBay) to change between dip tuning via a power meter and EPR recording via the spectrometer’s signal channel. The RF switch is manually operated by means of a power supply (on-off).

The power meter is an NI USB-5681 (ca 7000 €) from National Instruments (Austin, TX, USA). It operates from 10 MHz to 18 GHz with input up to +20 dBm. It has a maximum damage level of +30 dBm, which should well exceed the maximum output level of the RF amplifier. The power meter is the most expensive item on our shopping list. Other vendors carry similar meters for comparable prices. In principle, the dip tuning could also be performed, in a much cheaper way, on the voltage output of the detection diode (below); however, each diode should then be calibrated (dBm; voltage out) at regular intervals with a power meter. Also, the power meter can be attached to the output of any RF component in the bridge, and thus may prove to be indispensable for identifying a malfunctioning component.

The purpose of the diode is to detect the microwave signal reflected from the resonator and then amplified, and to convert it into a 100 kHz video signal that can be fed into the signal channel for demodulation, and whose average amplitude can be monitored by the PC for frequency-dip tuning. I use a PE 8013 (ca 1000 €) from Pasternack Enterprises (Irvine, CA, USA), which is a zero-bias Schottky detector with flat response over its 10 MHz to 18.5 GHz frequency range. Also, its transfer function for voltage versus incident microwave power is linear over the power range of our interest (0–20 dBm) and quasi-linear at lower values (cf Figure S3 in [5]). Its response compares favorably with that of other diodes that I have tried, and its stability in my hands is also better than tunnel diode detectors. Other vendors may, however, carry items of similar specifications. Note that the diode may be destroyed by application of excessive power. The maximum working power for the PE 8013 is 20 dBm and the maximally allowed peak input is 27 dBm, which is why we chose an amplifier that saturates around 20 dBm.

All RF components can be connected with short pieces (10–20 cm) of male-male RG402 coax cable except for the connection between the circulator and the resonator cell inside the magnet, which may vary depending on the positioning of the broadband bridge, and, especially, on the employed frequency. The connection may be made as short as possible (say 30 cm) to minimize losses at the highest frequencies; it can be made longer to increase the number of frequency dips (see Section 3.4) at lower frequencies, up to a few meters around 1 GHz, and tens of meters towards 0.1 GHz (cf [4]). An overview picture of the actual components is given in Figure 3, which also specifies a few required adaptors/connectors. I purchased all these connection items via eBay mainly from vendors in the PRC at an overall cost below 100 €.

### 3.3. Details of Home-Made Parts

A description of the resonator/sample holder has been given before and is reproduced here for convenience (Figure 4). The basic ingredients are always the same: a metal base plate of 1 mm thickness (not a critical dimension) is cut from a copper or aluminium plate. A female SMA connector is soldered to one end (the end of the aluminium cell is surrounded by a piece of 25 μm copper foil to allow soldering). The cell is insulated with 4–5 windings of 50 μm yellow acrylate tape of 5 cm width; this also fixes the copper foil onto the aluminium plate. Tape of other, especially darker, colors may cause background signals from, e.g., radicals and manganese. A rectangular cell compartment (sample compartment) is cut out of the tape, e.g., with a snap-off blade knife, down to the bare metal with dimension along the long cell axis that fits into the homogeneous field of the modulation coils. For doubling of the EPR signal amplitude, a mirror sample compartment may be cut out on the other side of the cell, whereby powder samples may later be held in place with, e.g., Parafilm wrapping. Then, a 0.2–0.25 mm wire of bare copper, or lacquered copper, or bare silver, is soldered to the inner conductor of the SMA connector and led over the cell (fixed in place with stripes of acrylic tape) to the sample compartment. It is then hand-wound around the sample compartment such that adjacent windings do not touch (separation space approximately equal to the wire diameter). From the end of the sample compartment, the wire is fixed with acrylic tape with a short lead towards the end of the cell. At no point are the wire and the base plate in electric contact. They form a microwave transfer line with full reflection at the open end. Note that for metalloproteins, aluminium is preferred over copper for the ground plate, and silver is preferred over copper for the wire, because aqueous protein solutions (that is, before the freezing act) have a significant tendency to liberate copper ions from solid copper, which in the spectroscopy results in Cu(II) background signals [5].

Figure 5 shows 2D drawings of resonators in three different forms. The simple, small cell (6.5 mm wide; sample compartment 4 × 16 mm^2^) is intended for setup testing, calibrating, familiarizing purposes. Its dimensions are limited such that it can be placed inside a regular X-band cavity so that the original coils integrated in the side walls of the cavity can be used for field modulation, and no home-made modulation coil assembly is required. Note that in this approach, the cavity is simply a housing and is not used for microwave storage. The intermediate-size cell in Figure 5 is the standard research cell for low-temperature work. It fits into the flow dewar, which in turn fits into the home-made modulation-coils assembly. The wide cell in Figure 5 is to optimize the EPR signal from powders of maximized sample size at room temperature when no flow dewar is required. Note that these cells were not designed to accept unstable samples. Nitrogen cooling of cells (cf [4]) holding aqueous-solution samples in a glove box will provide protection against denaturation by air.

In the last described version of the broadband machine, I used a ready cylindrical modulation-coils assembly that was taken from a Varian E-line Q-band spectrometer. My choice was based on the fact that it fitted an existing helium-flow dewar for the Q-band setup, to which the dimensions of the low-temperature broadband resonator were adjusted [11]. Obviously, these items are not generally available, hence the need to construct similar structures in-house. A picture of the home-made modulation-coils assembly is in Figure 6. The coils were obtained as brandless mobile phone inductive charging units (20 W) whose regulatory printed circuit boards were disconnected and discarded. The coil wire (12 turns) is of the Litz type made of 100 enameled thin copper wires each of 0.09 mm diameter. The coils were wire-connected in series and mirror-image folded over a plastic (Teflon) cylinder, with an inner diameter of ca 40 mm, such that their individual fields will add. When you are sure you have the right orientation, the coils can be provisionally attached to the cylinder with transparent tape. Sturdiness is then provided by surrounding the coils with heat shrink tubing of proper size. The leads to the coil pair should be connected to a twin BNC to double wire jack (Figure 6), so that it can be connected to the twin BCN male plug of the spectrometer’s modulation cable. The quality factor of the coil pair should be optimized for use with 100 kHz modulation. To this goal, measure the coils’ impedance, *L* (in Henry), here: 17 μH, with an impedance meter, or a multimeter with impedance option, and use the following well-known tank circuit resonance equation to calculate a compensating capacitance, *C* (in Farad), for a modulation frequency *ν* (in Hertz), where typically *ν* = 100 kHz,
(6)$$LC={\left(\frac{1}{2\pi \nu }\right)}^{2}$$
which should be added in parallel to the coils circuit in the form of a polypropylene film capacitor or a ceramic disc capacitor (do not use polarized capacitors). Then, fix the assembly in the gap of the magnet of the EPR spectrometer, e.g., with two pieces of polyurethane. For studies at ambient temperatures, the modulation cylinder may be placed horizontally inside the magnet; in combination with flow cryogenics, the orientation will generally be vertical. The spectrometer’s internal signal channel calibration procedure can now be employed to calibrate the coils using a BDPA sample (see Section 3.6). For the coils in Figure 6, this gives a maximum modulation amplitude of 13.5 gauss at 100 kHz. Modulation-field homogeneity was tested with two cells of type B (Figure 5), one of which had a spot sample of BDPA in the center of the sample compartment, and the other had five spots of BDPA in an X-form over the sample compartment. Both cells gave a single-line spectrum with peak-to-peak width of 0.7 gauss at 462 MHz.

A simple flow dewar (colloquially known as ‘the Swedish system’) for Q-band is described in [11], which was designed for magnets on a stand accepting a 30–50 L helium vessel underneath (as used in our lab). This simple dewar is topped with rubber corks and a plastic (PVC) elongation tube in which the low-T cell of Figure 5 can have its sample compartment cooled by helium flow, while its thin top extension is subject to a heat gradient towards ambient temperature. This system was used in my previous broadband EPR studies [4,5,6]. Unfortunately, there is no universal solution for the adaptation to broadband EPR of the various flow cryostats in use in different laboratories. Perhaps the most commonly employed systems are those from Oxford Instruments (Abingdon, UK). Their CF935P cryostat for Q-band cavities has an inner diameter of 43 mm, which is more than enough to accept our standard low-temperature cell (Figure 5B), or even the oversized cell (Figure 5C). Its closing flange (KF50) would allow for the construct of a simple adapter to accommodate broadband cells. Modification of their ESR900 cryostat for X-band cavities would require a rather more challenging replacement of its quartz dewar by an elongated glass dewar with a wide top.

### 3.4. Tuning Procedure

A user interface for the dedicated PC, written in LabVIEW, is employed for tuning the broadband bridge. The program can be found in Appendix A. A screenshot is given in Figure 7. After start-up, the program asks for the limits of a frequency range to be explored, which is then scanned at a low incident microwave power of −10 dBm, providing a pattern of potential frequency dips as exemplified in Figure 7. The user should then bracket any one of the dips, using a cursor, and re-scan (Figure 7). The chosen dip should preferably have its minimum below the input power. The frequency limits of the dip should then be refined to zoom in on a depth of ca 4 dB, and the input power should be raised such that the dip minimum corresponds to a power reading of ca 0–10 dBm (Figure 7). The source should now be set to radiate continuously at the frequency of the dip minimum and at the corresponding input power (‘operate’). The RF switch can now be engaged to route the reflected microwave to the detection diode. 

The spectrometer can now be run with its proprietary software in which the X-band bridge is not engaged (that is X-band tuning is at ‘stand by’). The spectrum is stored in the standard way, e.g., the resulting file(s) are Bruker .DSC (=description in ascii) and .DTA (=data in binary) files. The files are identical in every respect to files generated with the X-band bridge engaged except that the microwave frequency value in the .DSC file is zero. To fix this, open the DSC file, e.g., with Windows Notepad, and replace ‘MWFQ 0.0’ with the actually used frequency in Hz in Bruker notation, e.g., MWFQ 9.123456e+09.

### 3.5. Small-Sample Testing: Mn^2+^ in CaO

Employing the straight, mini test resonator (see Figure 5A), the broadband spectrometer can be tested without the need to build a modulation coil assembly, that is, by using the X-band cavity resonator, not as a resonator, but only as a broadband-cell holder with modulation coils in its side walls. The purpose of this setup is to provide the operator with a convenient way to check the success of the spectrometer conversion and to become familiar with the use of the instrument. It can also be used for magnet field and scan calibration purposes. The setup is limited in its sensitivity by the limited sample size (16 × 4 × 0.15 mm^2^) of the resonator, and it does not allow for cryogenic measurements. An illustrative example is given in Figure 8 using an Mn contamination in CaO. The commercial CaO sample has a quoted metal-based purity of 99.95%, therefore the Mn contamination is <0.05%. Manganese is 100% ^55^Mn with electron spin *S* = 5/2 and a nuclear spin *I* = 5/2, which splits the main *m_S_* = |±1/2> line from substitutional Mn^2+^ in the cubic CaO lattice into a six-line pattern. The combination of a small line width (i.e., strong signal) and relatively low Mn concentration (i.e., weak signal) affords a system that poses a mild challenge in terms of attainable signal-to-noise ratio. The individual spectra of Figure 8 were obtained as 16 × 100 s scans except for the lowest two frequencies, which took 36 × 100 s. The figure is a clear illustration of the reduction in sensitivity that comes with decreasing the microwave frequency as a result of a reduced Boltzmann population difference over the electron spin levels.

We now zoom in on the details of the spectrum taken at two significantly different frequencies of 17.8 and 3.9 GHz (Figure 9). The Mn^2+^ contaminant is substitutionally replacing Ca^2+^ in the cubic CaO lattice [12], so the spin Hamiltonian, with *S* = 5/2 and *I* = 5/2, is
(7)$$H=g\beta BS+ASI+\frac{1}{6}a{[S}_{x}^{4}+S_{y}^{4}+S_{z}^{4}-\frac{1}{5}S(S+1)(3S^{2}+3S-1)]$$
describing the electronic Zeeman interaction, the central hyperfine interaction, and the fourth-order cubic zero-field interaction [13]. Typically, *a* is much smaller than *A* [12], so the spectrum should consist of six hyperfine lines, whereby each line should be split into a symmetrical pentad from zero-field interaction. This latter fine structure is not observed (Figure 9), which presumably means that four of the five lines are broadened, due to a distribution in zero-field interaction strength. On the other hand, each central line (|+1/2> ←→ |−1/2>) is unaffected by the magnitude of *a*. Observed single satellite lines (Figure 9) are symmetrically separated about the center (≈*g* value) of the spectrum, and they are presumably from a slightly different Mn^2+^ site in these microcrystals of small dimensions. The satellite lines are frequently observed, but not commented on, in ‘powder’ samples of Mn^2+^ in CaO (e.g., [14]). 

Simulations of the transition within the *m_S_* = |±1/2> doublet (*A* = 85.7 gauss; *a* is undetermined) illustrate spectral-shape dependence on frequency. At 17.8 GHz, the hyperfine lines are almost equidistantly separated; the splittings at lowest and highest field differ by 5% only. In other words, the hyperfine interaction is close to a first-order perturbation of the Zeeman interaction. Contrarily, at 3.86 GHz the difference is increased to 21%. The hyperfine splitting pattern has clearly become asymmetric. The *g* value obtained from simulation is *g* = 2.0023 at 17.8 GHz, *g* = 2.0010 at 9.76 GHz (not shown), and *g* = 1.9968 at 3.86 GHz, and since there is no reason why the *g* value should be frequency-dependent, this suggests that the fields near *g* ≈ 2 at the extreme frequencies require calibration correction (see below). 

### 3.6. Field Calibration with DPPH and BDPA

Polycrystalline DPPH (2,2-diphenyl-1-picrylhydrazyl) is an indefinitely stable *S* = 1/2 radical with a single-line spectrum in the X-band (*g* = 2.0036) of circa 2–3 gauss peak-to-peak linewidth from exchange narrowing and therefore high signal-to-noise ratio, which has been widely and extensively used as a field marker and sensitivity marker. Its use in high-frequency EPR has been under debate where several authors report asymmetry or resolution of spectral structure at higher frequencies [15,16,17,18] while others find a single line of width proportional to the frequency [19]. Differences in *g* anisotropy may be related to different synthetic routes [20]. Due to its strong signal, DPPH is easily measurable with a mini test resonator in the present setup. Using a sample recently obtained from Sigma-Aldrich, I measured the spectrum at frequencies near the extremes of the available range (Figure 10 top). Here, broadband EPR proves its merit for deconvolution of frequency-independent and frequency-dependent interactions. 

At 102 MHz, the spectrum is a single line of 2.2 gauss width, consistent with earlier work down to a few MHz [21]. Contrarily, at 17.9 GHz, spectral anisotropy starts to resolve, which implies that its reliability as field marker will worsen with increasing frequency. Following field calibration at the X-band, the theoretical resonance field for *g* = 2.0036 at 102 MHz of 36.3 gauss is found experimentally to be 41.0, that is, off by +4.7 gauss. The corresponding field at 17.9 GHz is 6382.6 gauss, and the experimental field can only be estimated to be off by ca −3 to 4 gauss. Calibrations at higher frequencies will be increasingly uncertain. Recall that previously one high-frequency study of DPPH from the same manufacturer did not resolve anisotropy [19], so reproducible results over different batches are apparently not guaranteed.

A possible alternative for DPPH is the polycrystalline, stable *S* = 1/2 radical BDPA (α,γ-bisdiphenylene-β-phenylallyl) complex with benzene (1:1), which exhibits a single-line spectrum in the X-band (*g* = 2.00254) with even smaller peak-to-peak linewidth of circa 0.6–0.7 gauss, and which is recommended by EPR spectrometer manufacturer Bruker for modulation frequency calibration purposes [14]. For this compound, no anisotropy is resolved (Figure 10 bottom), which suggests that BDPA (at least from this batch) is an appropriate field calibration marker for broadband EPR over the available frequency range. The field deviations of the EMXplus 9 kgauss magnet, found with BDPA, are +4.7 gauss at 102 MHz and −3.6 gauss at 17.9 GHz.

## 4. Discussions

Over the last decade, I have developed a broadband continuous-wave EPR spectrometer for the approximate frequency range of 0.1–18 GHz. In the present paper, the building of the instrument has been simplified to the implementation of a conversion kit into existing X-band spectrometers. This leads to a considerable reduction in construction costs and complexity. The elements of the kit consist of readily purchasable parts plus a few items that have to be manufactured in-house. Fabrication of a simple mini cell is the only requirement for initial testing and getting acquainted with the setup. The complete kit is universally applicable, except for the adaptation of the cryogenics, whose geometry will depend on the targeted X-band spectrometer. Operation of the modified spectrometer requires only minor additional instruction.

In terms of application, the reader is advised that the broadband spectrometer is not intended to replace single-mode resonator spectrometers. Application of the described extension should rather be seen as a natural follow-up to initial studies with the standard X-band spectrometer. The latter is superior in sensitivity, although this may not necessarily be true for cavity spectrometers at other frequencies. Also, attainable energy density in the broadband resonator is less than in X-band cavities with their high-quality factors, which makes it more difficult to saturate signals. Thus, if power saturation characteristics are desired, they should be determined in the standard spectrometer. Also, since operation frequency and dip characteristics may depend on exact cell geometry and sample permittivity, quantitation (that is, spin counting) versus a standard compound of known concentration is more readily achieved with the standard spectrometer. The broadband extension is specifically intended for spectral analysis from systems with all but the simplest spin Hamiltonians. A broadband data set provides a much more rigorous test for interpretation of the details of spectral shapes and line-broadening mechanisms.

## 5. Materials and Methods

DPPH (2,2-diphenyl-1-picrylhydrazyl), and BDPA (α,γ-bisdiphenylene-β-phenylallyl) complex with benzene (1:1) were obtained from Sigma-Aldrich of Merck (Amsterdam, The Netherlands). Mn:CaO was obtained from Alfa Aesar of Fisher Scientific (Landsmeer, The Netherlands) as calcium oxide 99.95% (metals basis). Silver wire was from Alfa Aesar as soft, annealed silver wire, 0.25 mm in diameter, 99.9% (metals basis). The standard X-band spectrometer is a Bruker EMXplus (Bruker Physik AG, Karlsruhe, Germany). All hardware for its conversion into a broadband spectrometer is detailed in Section 3. Used software was written in LabVIEW (2020) and is given in Appendix A. For proper operation, the BB-Tuner program requires (free) downloading from the NI site and installing of the driver for the NI USB-5681 power meter. The Z-Calculator, for resonator design, does not require additional software.

## Figures and Tables

**Figure 1 molecules-28-05281-f001:**
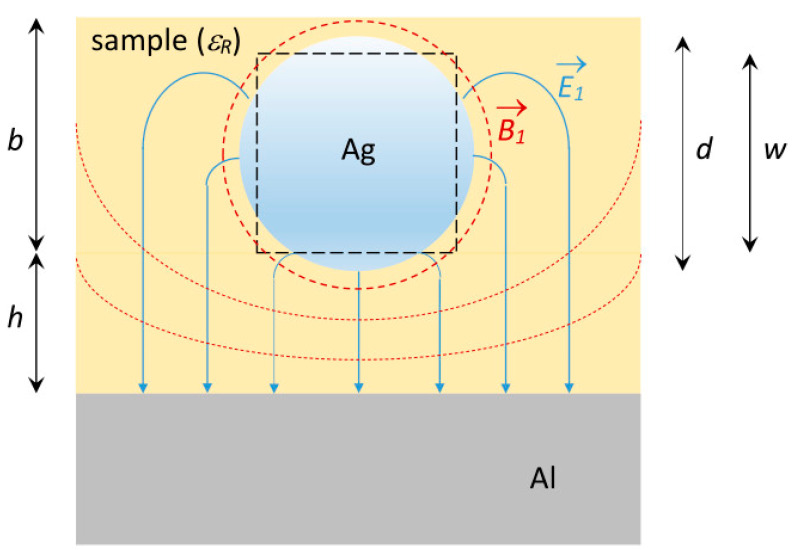
Wire micro strip transmission line for EPR spectroscopy. In this schematic drawing of a cross section through a transmission line, the diamagnetic isolation between a silver wire of diameter *d* and an aluminium ground plate has been replaced with a paramagnetic sample of height *h* + *b* and with electric permittivity *ε_R_*. The square (black broken line), with side length *w*, is the surface equivalent of the cross-sectional circle of the silver wire. *E*_1_ and *B*_1_ are the vectorial field lines of the electric and magnetic components of the transmitted microwave. The resonator and sample-holder cell to which this cross section belongs is placed in the spectrometer such that the static external magnetic field vector, *B*_0_, is perpendicular to the plain of the drawing, so that all *B*_1_ lines are perpendicular to the *B*_0_ field, as required for EPR. The parameters *d* (or *w*), *h*, *b*, and *ε_R_* are optimized to obtain a line characteristic impedance *Z*_0_ ≈ 50 Ω.

**Figure 2 molecules-28-05281-f002:**
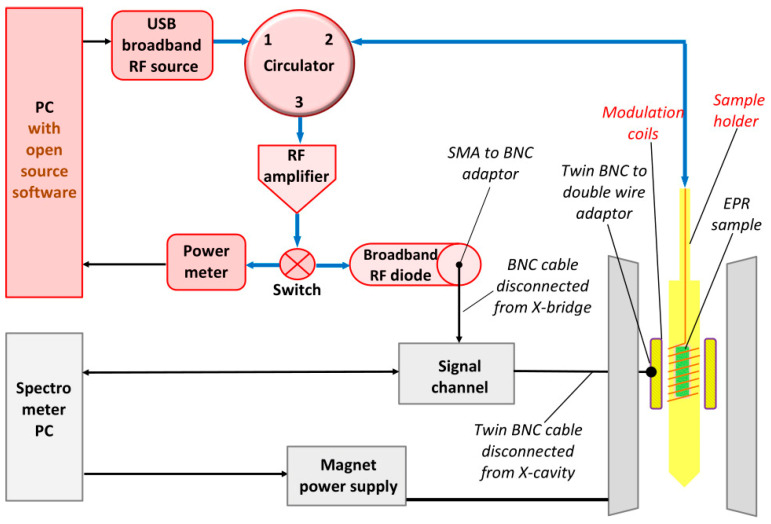
Schematic overview of the proposed broadband EPR spectrometer. An existing, single-frequency, X-band CW spectrometer (parts in gray) with its microwave bridge (not shown) disabled and its cavity removed is extended with a conversion kit consisting of commercially available parts (in pink) and home-made parts (in yellow). Required signal adaptors are also indicated. For clarity, a cooling system is omitted from the drawing.

**Figure 3 molecules-28-05281-f003:**
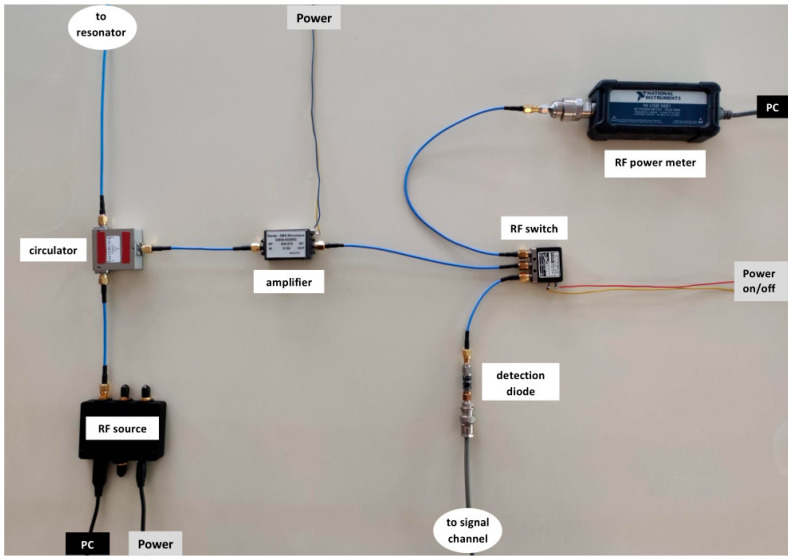
Photograph of the actual components of the broadband microwave bridge. The RF source and RF amplifier are active components and should be protected from overheating, e.g., by attaching to an aluminium plate (here omitted). ‘Power’ is a connection to a low-voltage power supply. ‘PC’ is a connection to a dedicated computer. The shortest coax cable, e.g., between circulator and amplifier, is 10 cm. Electrical grounding of hand wrists is strongly recommended before assembly.

**Figure 4 molecules-28-05281-f004:**
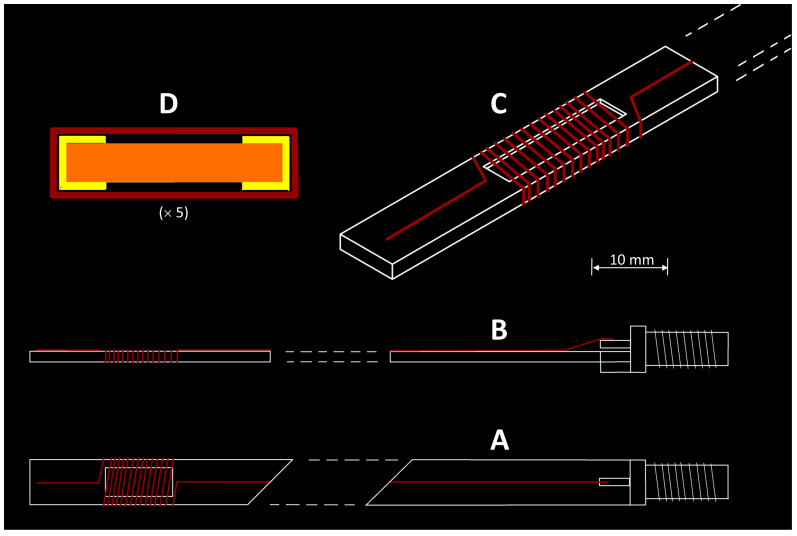
Isometric drawing of a wire micro strip cell. (**A**) Top view: (**B**) side view; (**C**) 3D view; (**D**) cross view. The cross view shows acrylic tape isolator (yellow) with sample compartments on both sides (black) surrounded by a low-permittivity wrap (brown), e.g., parafilm, or an extra layer of acrylic tape, to keep the sample in place. See the main text for further details. This figure has previously been given as Figure S1 in the Supporting Information to [4].

**Figure 5 molecules-28-05281-f005:**
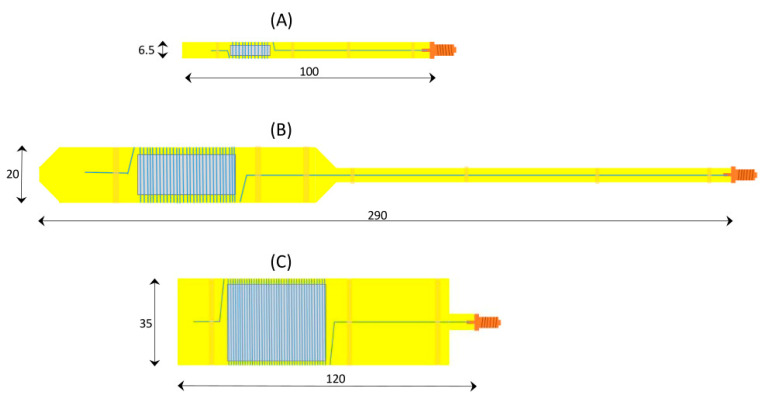
Geometries of the resonator cell intended for different applications. (**A**) Mini cell with small sample compartment and with dimensions that allow its placement in a rectangular X-band cavity, intended for learning and calibration purposes; (**B**) standard cell whose sample compartment fits within the homogeneous-field space of the dedicated modulation-amplitude coils, and whose dimensions allow it to be used in combination with a dedicated He/N_2_ flow dewar; (**C**) oversized cell with maximized sample compartment for use at ambient temperature.

**Figure 6 molecules-28-05281-f006:**
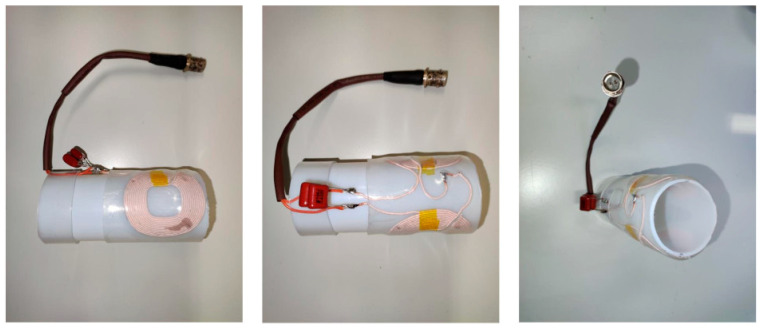
Photograph of modulation-coils assembly. The construct is based on a plastic cylinder on which two Litz-wire coils, taken from cell phone chargers and connected in series, are folded and attached. The diameter of the cylinder is such that it accommodates either room-temperature cell C of Figure 5 or a cooling system in combination with cell B of Figure 5. Note the film capacitors, with C = 100 + 68 nF, that tune the tank circuit to resonance at 100 kHz, and the twin BNC jack for connection to the spectrometer’s modulation cable.

**Figure 7 molecules-28-05281-f007:**
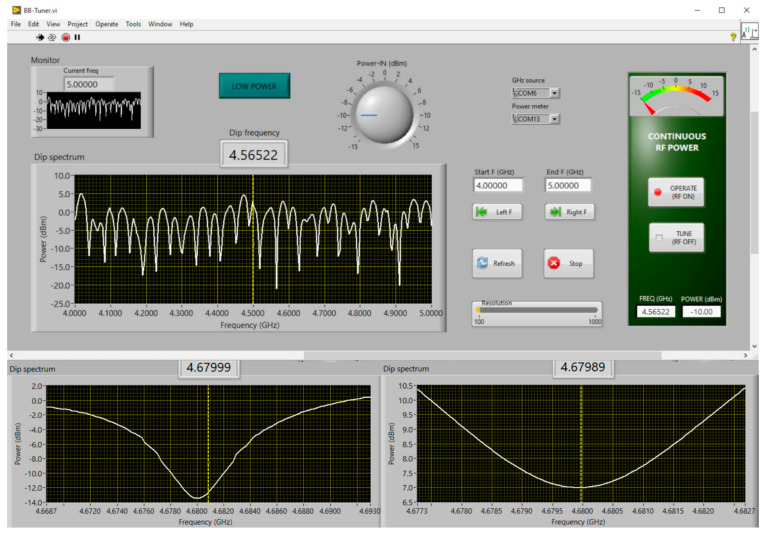
Screen shot of the dip-tuning program. The LabVIEW program monitors the signal from the power meter as a function of the scanned microwave frequency. (**Upper panel**) After an initial choice of frequency limits (here: 4–5 GHz), the program produces the power response at −10 dBm input power. A cursor is then used to single out an individual dip, using the ‘left’ and ‘right’ buttons; (**lower left panel**) this dip peak is re-scanned at low power and is then further reduced, using the cursor, to limiting power levels of circa 4 dB above the dip power; (**lower right panel**) input power is raised such that a dip power ensues that will be optimal for the detection diode, that is, circa 0–10 dBm. The bridge is now tuned and can be turned to permanent RF output (‘operate’), and reflected power can be switched, with the SPDT switch, to the detection diode for EPR spectroscopy.

**Figure 8 molecules-28-05281-f008:**
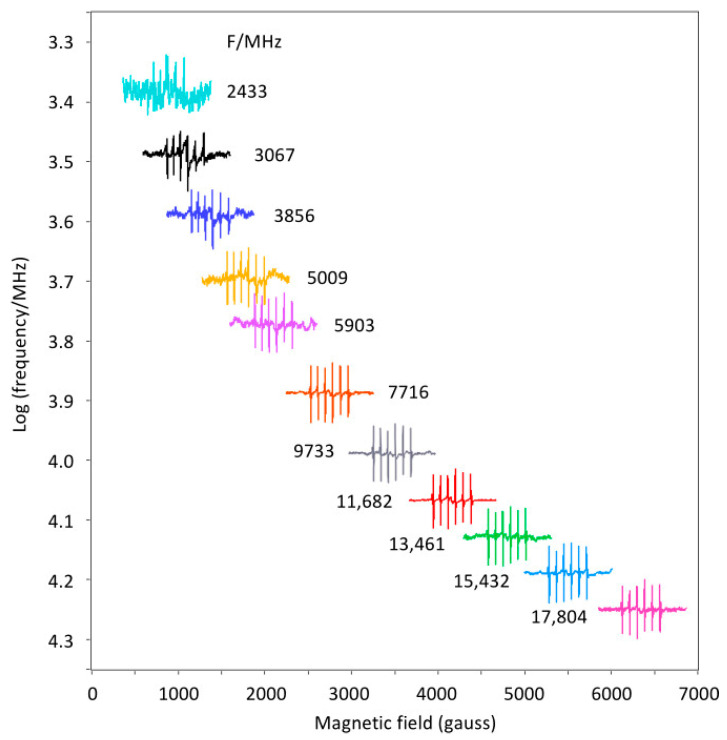
Broadband EPR of Mn^2+^ trace contaminant in cubic CaO. The room-temperature spectra were obtained at the indicated frequencies with the low-sensitivity mini cell A of Figure 5 and with three different circulators (2–6, 7–16, and 11–18 GHz). The 100 kHz modulation amplitude was 1 gauss. The figure illustrates the field range required for broadband EPR up to circa 18 GHz, and also the decrease of signal-to-noise ratio with decreasing frequency.

**Figure 9 molecules-28-05281-f009:**
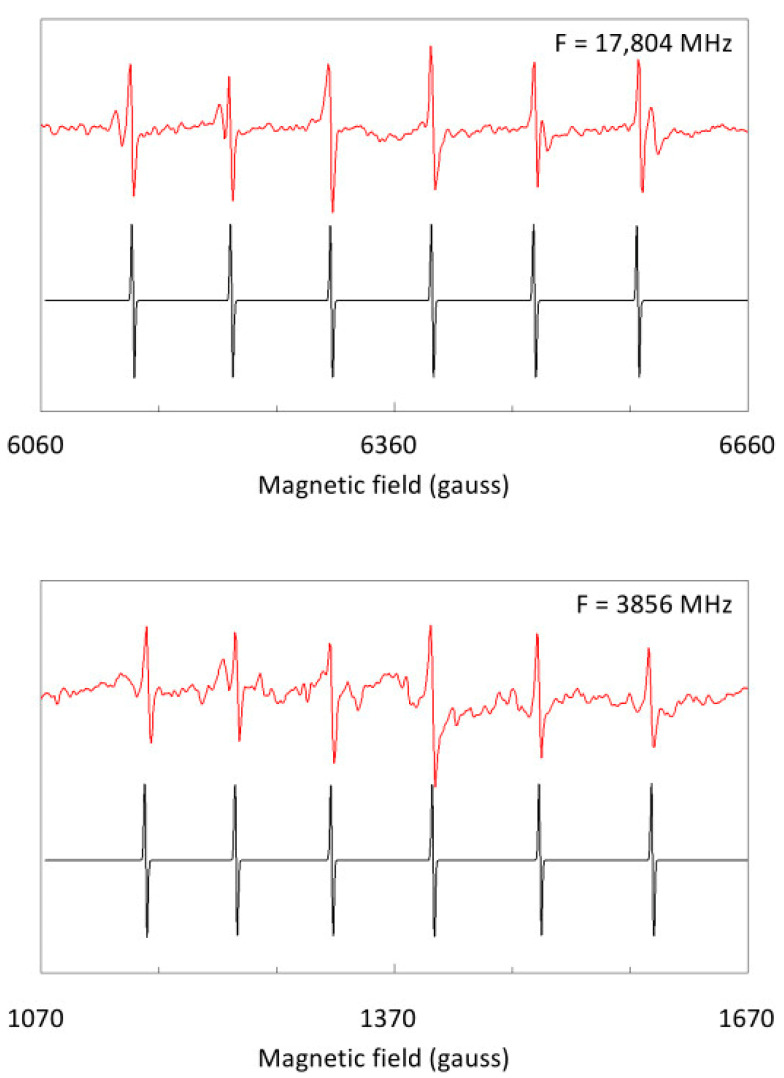
Comparison of hyperfine details from Mn^2+^ in CaO at 17.8 versus 3.9 GHz. The 100 kHz modulation amplitude was 1 gauss. The spectra at 17,804 and 3856 MHz are compared to identify frequency-dependent changes. With the help of simulations of the main signal from the transition within the *m_S_* = |±1/2> doublet, a deviation from field-equidistant hyperfine splittings is found, which increases with decreasing frequency.

**Figure 10 molecules-28-05281-f010:**
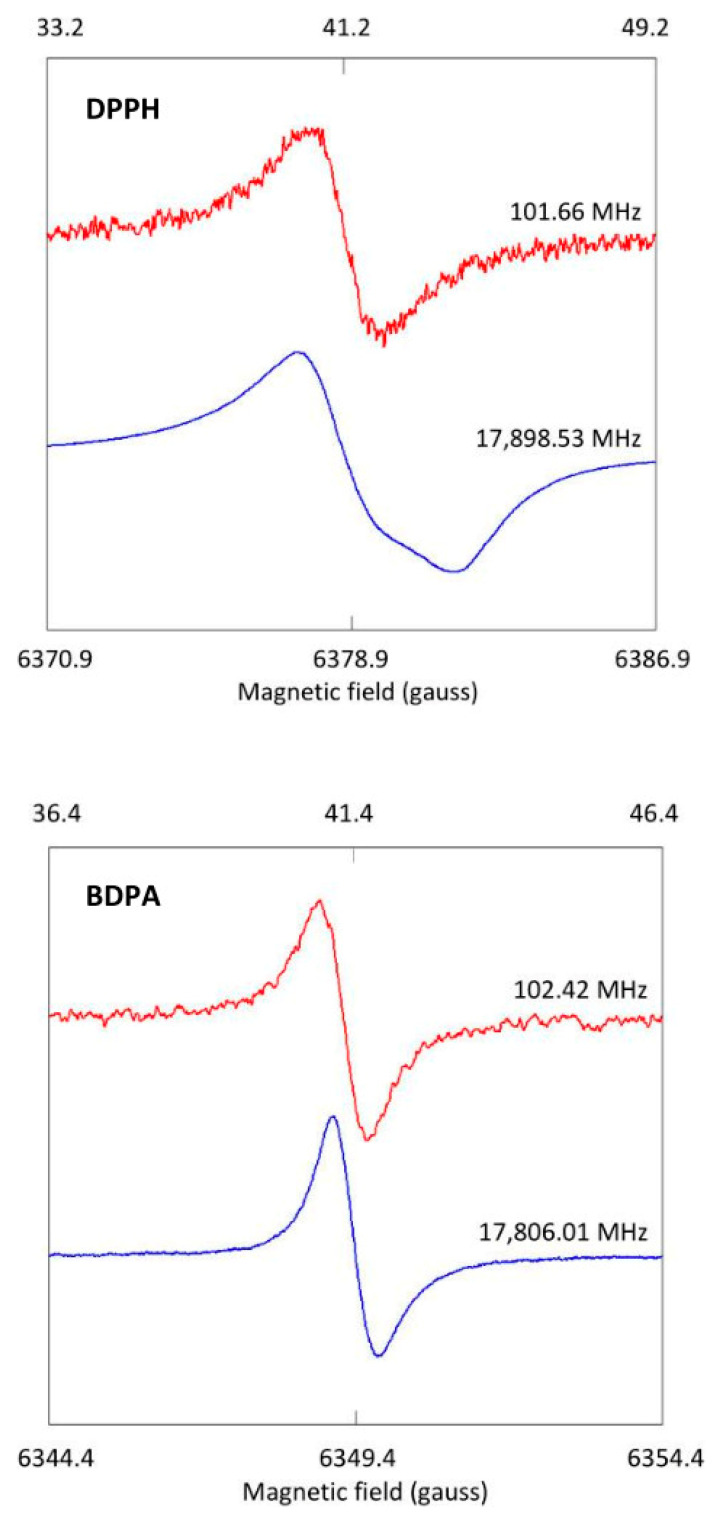
Frequency dependence of DPPH and BDPA calibration probes. DPPH (**top**) and BDPA (**bottom**) were measured in the mini cell of Figure 5A at two extreme microwave frequencies. The spectrum of DPPH can be seen to develop anisotropy (resolution of *g* matrix) at 17.9 GHz, while the spectrum of BDPA remains essentially isotropic over the full frequency range. BDPA is thus a suitable probe for accurate field calibration in broadband EPR, and DPPH is not.

## Data Availability

The data are contained within the article.

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
