# Peer review of "Conversion of a Single-Frequency X-Band EPR Spectrometer into a Broadband Multi-Frequency 0.1–18 GHz Instrument for Analysis of Complex Molecular Spin Hamiltonians"

_molecules, 2023, doi:10.3390/molecules28135281_

Round 1

Reviewer 1 Report

The manuscript reports the Conversion of a single-frequency X-band EPR spectrometer into a broadband multi-frequency 0.1-18 GHz instrument. Detailed description of the conversion is presented and the sample testing proves that the multi-frequency spectrometer works well.  The overall research is systematically done on a good level and the conclusions are convincing. This paper can be accepted by Molecules after appropriately replying the following question.

In section 4.3, it is very hard to link the text to Figure 4.  The author could improve this section to make it easier for reader to follow the text. For example, change Figure 4 into a 3D one.

Author Response

I fully agree with the reviewer; thank you for the suggestion. The basic outline of the resonator has been previously given in a 3D drawing, however, in the supporting information to ref [4]. Since that material is not under copy right, I reproduce the figure here in the main text to improve readability of section 4.3.

Reviewer 2 Report

The long title adequately describes the scope of this paper, which is a continuation of a series of efforts by Dr. Hagen to create EPR spectrometers at a wide range of microwave frequencies.  His prior papers are references 3-6.

This is a very conversational paper – written as if the author were describing the spectrometer to a visitor to his lab.  The result is that the reader is frustrated by the lack of crucial information needed to reproduce the spectrometer.

Where in the Bruker bridge, for example, was the output of the RF diode connected?

Were the modulation coils driven by the Bruker modulation driver so that the EPR signal was detected via the Bruker 100 kHz lock-in amplifier system?

The coaxial cable used has very high transmission loss above a couple GHz.  Are the long lines needed to create the pattern of “dips” shown in figure 6?  The text is not very clear about frequency selection, but it seems that one simply selects a frequency at which the reflection signal as shown in figure 6 is a minimum.  If this is correct, (or especially if it is not!) the frequency selection method and the tradeoffs regarding frequency selection should be described in more detail.

There is no information about absolute sensitivity at any frequency, although prior papers in the series have acknowledged that the previous incarnations were not very sensitive.  The manufacturer’s specifications for the microwave source used immediately tell one that the spectrometer is probably orders of magnitude less sensitive than the Bruker EMX that it is designed to supplement.

Section 4.5 is titled small-sample testing.  How many spins were measured?  Was the modulation amplitude and incident power selected to be non-distorting and non-saturating?

The description of the modulation coils (figure 5 and associated text) is mind-boggling.  The author did not even tell what model of cell phone charger was torn apart to get the coils.  There coils and matching them to the Bruker console are one of the most crucial aspects of reproducibility of the system described.  The paper must say how many turns of what type Litz wire was used.  What was the impedance when measured at 100 kHz?   Most modulation coils are actually 4-coil sets to provide greater field homogeneity.  The photograph and description seem to say that this was only a 2-coil set.  Was the field homogeneity measured?

The resonators are better described in reference 4, which actually shows photographs in addition to diagrams and calculations.

The resonators are designed for strong samples and for samples that do not have to be evacuated or otherwise protected from the ambient air.  This should be explicitly stated.

The introduction mentions that low-frequency EPR spectrometers “have been available previously.”  This statement could be substantiated by citing the chapter on low-frequency spectrometers in Biol. Magn. Reson. volume 21.

At the top of page 2 the word divulgence might be replaced by distribution.

These comments and questions are merely indicative of the extensive changes required to make the paper publishable.

except as mentioned the English is OK

Round 2

Reviewer 2 Report

The author's response is almost comical, saying that enough information was given to guide others to construct an instrument, but also that the paper was not for those who build their own instruments.  Sometimes arrogant dismissal of a reviewer contradicts itself.

The reviewer accepts that the author's style is his choice.  Sufficient changes were made to publish the paper as is.  Readers can ask the author for needed information.